# Salivary gland ultrasound is associated with the presence of autoantibodies in patients with Sjögren's syndrome: A Danish single-centre study

**Nanna Surlemont Schmidt**[1,2☯], **Anne Voss**[1,3☯], **Anna Christine Nilsson**[3,4], **Lene Terslev**[5,6], **Søren Andreas Just**[1,7], **Hanne M. Lindegaard**[1,3☯]*

**1** Department of Rheumatology, Odense University Hospital, Odense, Denmark, **2** OPEN, Open Patient data Explorative Network, Odense University Hospital, Region of Southern Denmark, Odense, Denmark, **3** Department of Clinical Research, University of Southern Denmark, Odense, Denmark, **4** Department of Clinical Immunology, Odense University Hospital, Odense, Denmark, **5** Copenhagen Center for Arthritis Research, Center for Rheumatology and Spine Diseases, Rigshospitalet Glostrup, Glostrup, Denmark, **6** Department of Clinical Medicine, University of Copenhagen, Copenhagen, Denmark, **7** Section of Rheumatology, Department of Medicine, Svendborg Hospital–Odense University Hospital, Svendborg, Denmark

☯ These authors contributed equally to this work.
* Hanne.lindegaard@rsyd.dk

**Data Availability Statement:** All relevant data are within the manuscript and its Supporting Information files.

## Abstract

### Objectives

To investigate whether ultrasound findings of major salivary glands are correlated with serological markers, autoantibodies, patient- or doctor-reported disease activity in a Danish cohort of patients with primary Sjögren's Syndrome (pSS).

### Methods

In all, 49 patients at Odense University Hospital with pSS diagnosed according to the 2002 American-European Consensus Group (AECG) classification criteria were included. Patients were characterized using the EULAR Sjögren's Syndrome Disease Activity Index (ESSDAI, score of systemic complications) and EULAR Sjögren's Syndrome Patient Reported Index (ESSPRI), serologic markers, Schirmer's test and salivary test. Salivary gland ultrasound (SGUS) was performed of the submandibular and parotid glands and scored according to the Outcome Measures in Rheumatoid Arthritis Clinical Trials (OMERACT) semi-quantitative scoring system.

### Results

More patients with abnormal SGUS had antinuclear antibodies (ANA) (p = 0.002), anti-Ro52 (p = 0.001), anti-Ro60 (p<0.001), anti-La (p<0.001) and IgM-RF (p<0.001). Titers for ANA (p = 0.02) and anti-Ro52 (p = 0.03) were higher in patients with abnormal SGUS. Twenty-three of the pSS patients had no pathological findings on SGUS. There was no correlation between SGUS severity and ESSDAI- or ESSPRI-scores.

**Funding:** The study was supported by the Danish Rheumatism Association. Grants R181-A6347 and R178-A6346. We wish to state that the funders had no role in study design, data collection and analysis, decision to publish, or preparation of the manuscript.

**Competing interests:** The authors have declared that no competing interests exist.

## Conclusions

Abnormal SGUS findings are associated with autoantibodies of high specificity for pSS but not with ESSDAI, ESSPRI or inflammatory markers.

## Introduction

Primary Sjögren's syndrome (pSS) is a chronic systemic autoimmune disease characterized by mucosal dryness, especially of the eyes and mouth, and extra-glandular manifestations as muscle and joint pain, interstitial lung disease and fatigue. Most patients have specific autoantibodies (anti-Ro and anti-La) and high levels of plasma immunoglobulins [1]. The pathogenetic impact of these findings have been studied extensively, and inflammation with B-cell activity in salivary gland tissue in pSS is thought to have pathogenetic significance via autoantibody synthesis and secondary cellular immune response [2]. Additionally, patients with pSS have an increased risk of developing non-Hodgkin lymphoma, and risk factors include recurrent swelling of the parotid glands, high EULAR Sjögren's Syndrome Disease Activity Index (ESSDAI) and cryoglobulinemia [1].

There are no diagnostic criteria for pSS, but in clinical practice, the diagnosis is based on the classification criteria provided by the American-European Consensus Group (AECG), proposed in 2002 and on the American College of Rheumatology (ACR) and the European League Against Rheumatism (EULAR) consensus criteria from 2016 [3, 4].

The salivary gland (SG) function and pathology can be characterized, as suggested in the 2002 AECG criteria, through minor SG biopsy quantifying a focus score (defined as infiltrates of mononuclear cells with at least 50 cells per 4 mm$^2$) and assessing unstimulated salivary flow. Further, methods to evaluate the SG include sialography, scintigraphy, computed tomography (CT) scan, magnetic resonance imaging (MRI) and ultrasound. Ultrasound of the salivary gland can assess the parenchymal changes related to pSS, which involve varying degrees of inhomogeneity of the parenchyma [5–7]. A modest correlation between salivary gland ultrasound (SGUS) findings and the focus score has been reported [7, 8].

Several studies have assessed the use of SGUS in routine care for diagnosing pSS, suggesting that SGUS improves the sensitivity of the diagnostic criteria with minimal impact on specificity [5–7]. Some studies have found associations between SGUS and the histopathology of SG, serology and clinical disease outcomes [9–11]. SGUS is a feasible method for assessing the parotid (PG) and submandibular glands (SMG) with an examination time of approximately 10 minutes.

In the literature, several definitions and scoring systems have been proposed for assessing ultrasound changes in pSS patients, but the reliability has been varying [12–14]. Recently the ultrasound working group of the Outcome Measures in Rheumatology Clinical Trials (OMERACT) has developed and validated ultrasound definitions and a semi-quantitive scoring system (0–3) for elementary lesions in major salivary glands of patients with pSS with excellent intrareader reliability and good inter-reader reliability in video clips and in patients [15, 16].

The aim of this study was to investigate whether ultrasound findings of major salivary glands using the OMERACT definitions and scoring system are correlated with autoantibodies, patient- and doctor-reported disease activity in a Danish cohort of patients with pSS.

## Methods

### Study design and patients

Patients followed at Odense University Hospital with the diagnosis pSS based on the AECG criteria of 2002 and with appointments in the out patients clinic in the period November 2019

to February 2020 (108 patients) were invited to participate in the project. In all, 49 consecutive patients were included (45.4%) in a cross-sectional study. During the study visit, all participants were interviewed, including registration of current medication, and had an ultrasound of the major salivary glands, tear flow test, saliva flow test and blood sampling performed. No information about non-participants is available. The study protocol was approved by the Regional Committees on Health Research Ethics for Southern Denmark (project ID: S-20190066) and approved by the Regional Data Protection Agency (19/35694). All patients gave informed consent after verbal and written information according to the approval.

## Ultrasound

Grey-scale ultrasound of the PG and SMG was performed on Siemens® ACUSON Sequoia Ultrasound System (Erlangen, Germany) equipped with a 18L6 linear array transducer.

The PGs and SMGs were examined according to the EULAR recommendations [15]. The PG were assessed both in the longitudinal and transverse plane, and the SMGs were scanned in the longitudinal plane. The patient was lying in a supine position with the neck extended. The ultrasound exam took approximately 10 minutes and was performed by the same examiner (NSS). Representative images of the four SGUS images in each patient were recorded and stored for scoring using a four-grade scale from 0–3 based on the OMERACT scoring system where grade 0 represents normal parenchyma, grade 1 mild inhomogeneity and no anechoic/hypoechoic areas, grade 2 moderate inhomogeneity and focal anechoic/hypoechoic areas and grade 3 diffuse inhomogeneity with anechoic/hypoechoic areas occupying the entire gland or a fibrous gland Fig 1 [15, 16].

In the following study, a patient was classified as having pathological SGUS consistent with pSS if at least one of four salivary glands (PG or SMG) with an OMERACT grade 2 or 3 was found, in line with published data [17, 18]. Prior to the study, the ultrasonographer was trained in the OMERACT scoring system with good intra- reliability [19].

## Clinical and patient-reported outcomes

The systemic activity of pSS, was recorded according to the ESSDAI. A score from 0–123 was given from the 12 domains included in the ESSDAI with different weight representing

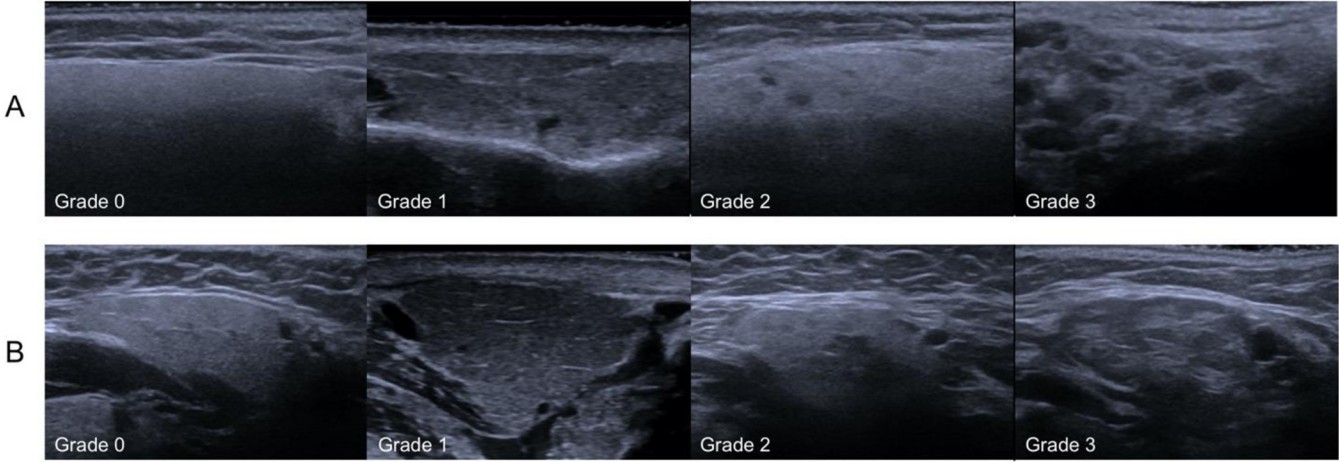

**Fig 1. Examples on OMERACT SGUS scoring system.** (A) Ultrasound images of parotid glands and (B) submandibular glands affected by pSS. Grade 0: Normal parenchyma, grade 1: Minimal change, mild inhomogeneity without anechoic/hypoechoic areas, grade 2: Moderate change, moderate inhomogeneity with focal anechoic/hypoechoic areas, grade 3: Severe change, diffuse inhomogeneity with anechoic/hypoechoic areas occupying the entire gland surface.

different aspects of systemic involvement: constitutional, lymphadenopathy, glandular, articular, cutaneous, pulmonary, renal, muscular, peripheral nervous system, central nervous system, haematological and biological domain [20]. The patients were divided in three groups: low activity (ESSDAI<5), moderate activity (5≤ESSDAI≤13) and high activity (ESSDAI≥14),

Patient-reported symptoms were recorded from the EULAR Sjögren's Syndrome Patient Reported Index (ESSPRI), including dryness, pain and fatigue reported on a VAS-scale (0–10) for each of the three symptoms [21].

## Saliva flow, tear flow and blood samples

An unstimulated saliva flow test was performed mid-morning, and the patients avoided smoking, chewing gum, eating and drinking at least two hours before testing. The patients were placed in an upright position collecting saliva in a beaker by drooling or gently spitting. The saliva was collected over 15 minutes and measured by a syringe. The saliva was expressed in millilitres (ml), and a positive test was perceived as volume less than 1.5 ml. Tear flow was measured by Schirmer's test. One filter paper strip was placed into the lower eyelid pouch of both eyes for 5 minutes. The test was considered positive if less than 5 mm of the filter paper were wet.

Blood samples were analyzed for C-reactive protein (CRP), complement C3c, C3d and C4c, cryoglobulin, and immunoglobulins (IgM, IgA, IgG). Antinuclear antibodies (ANA) were evaluated by indirect immunofluorescence (IIF) on HEp2 cells. Screening was performed in a 1:160 dilution, and positive samples were titrated to a maximum of 1:1280. Fluorescence pattern was reported according to International Consensus on ANA Patterns (ICAP) standards. Salivary gland antibodies were determined by IIF on primate salivary gland sections. Anti-double-stranded DNA (dsDNA) IgG was analyzed by enzyme linked immunosorbent assay (ELISA) and IIF on Crithidia Luciliae. Furthermore, samples were tested for IgM rheumatoid factor (IgM-RF) (ELISA), anti-Ro52 and anti-Ro60 (SSA), anti-La (SSB) the three last by chemiluminescence immunoassay (CLIA). All analyses of autoantibodies possibly related to pSS were performed in a tertiary care hospital laboratory accredited according to the EN: ISO 15189 standard.

## Statistical analysis

Continuous variables were expressed as mean ± standard deviation (SD) and categorical variables as numbers (percentages). Test for normal distribution was performed on all continuous variables.

Associations between and ESSDAI, ESSPRI, laboratory values, treatment and SGUS, were assessed with chi-square test, ANOVA and Kruskal-Wallis test as appropriate.

All statistical analysis was performed in Stata IC version 16.1.

## Results

### Demographics

The cohort consisted of 49 patients, all diagnosed according to the 2002 AECG criteria [1]. All patients also fulfilled the ACR/EULAR classification criteria of 2016 [2]. Of the 49 patients, 24 patients (49.0%) had undergone labial salivary gland biopsy, and 21 patients (42.9%) had focal lymphocytic sialoadenitis with a focus score >1 foci/4 mm$^2$. Demographic, laboratory and treatment characteristics are summarized in Table 1. Forty-seven out of 49 participants were female and the mean age of the cohort were 61.2 ± 10.1years. Three of the participants were current or former smokers and the mean BMI was 25.4 ± 5.6. Three patients had former non-Hodgkin's lymphomas and one was diagnosed with present non-Hodgkin's lymphoma.

Twenty-three patients (46.9%) had low, 18 (36.7%) had moderate and 8 (16.3%) had high disease activity measured as ESSDAI. Patients with high disease activity had a lower BMI than patients with low or moderate disease activity (p = 0.004).

## Biochemical assessments

Most patients were ANA positive by IIF on HEp2 cells (85.7%). Anti-Ro52 was detected in 63.3%, anti-Ro60 in 61.2% and anti-La in 46.9% of pSS patients. Nearly a quarter of the patients had anti-dsDNA and 2/3 were IgM-RF positive. The presence of these autoantibodies was not associated to one of the three ESSDAI groups (Table 1).

All patients had a normal level of CRP (<6 mg/L) and their levels of immunoglobulins A, G and M were all also within the normal range, even if there was a tendency to increasing levels of IgG with increasing ESSDAI, not statistical significant. There was an indication of complement activation as measured by elevated levels of C3d but levels of complement C3c as well C4c were in the normal range. Twentytwo percent of the patients had cryoglobulin, with no difference between ESSDAI groups (Table 1, p = 0.62).

## Treatment

Almost half of the patients were on hydroxychloroquine and 9 patients (18.4%) were treated with immunosuppressants such as prednisolone, azathioprine, mycophenolate mofetil and rituximab.

## ESSPRI

Self-reported symptoms as represented by ESSPRI were dominated by high levels of dryness and fatigue; nevertheless, ESSPRI scores were not associated with ESSDAI, neither in the subgroups nor in the total ESSPRI score (see Table 1).

## SGUS findings

The cohorts total average SGUS score was 5.7 (SD 4.5) but 23 (46.9%) of the patients had no glands with SGUS abnormalities defined as OMERACT grade 2 or 3, equivalent to nearly half of the patients with a well-established pSS diagnosis (Table 2). Of the 26 patients (53.1%) with SGUS abnormalities, 12 (24.5%) had at least one salivary gland with an OMERACT score of 3 distributed as 9 (18.4%) with at least one PG scored 3 and 10 (20.4%) with at least one SMG scored 3 (Table 2). All but one of the 26 patients (53.1%) with SGUS abnormalities in at least one salivary gland also had SGUS abnormalities in at least two salivary glands (51.0%) and 10 (20.4%) had at least two salivary glands scored 3 (Table 2). Of the 23 (46.9%) patients with no SGUS abnormalities 13 (26.5%) had a SGUS score of 1 in at least one salivary gland.

## Associations of SGUS and autoantibodies, Schirmer's/sialometry, ESSDAI and ESSPRI

When analyzing the patients according to pathology identified by SGUS findings in one or more glands (US+) versus no gland pathology (US-) (Table 3), significantly more patients with abnormal SGUS had positive ANA, anti-Ro52, anti-Ro60, anti-La and IgM-RF. The titer of ANA and anti-Ro52 were also higher in patients with abnormal SGUS. SGUS findings were not associated with ESSDAI- or ESSPRI-scores. Abnormal SGUS was not associated to abnormal Schirmer's test or abnormal sialometry; it is, however noticeable that most patients had abnormal function (67.8% had abnormal Schirmer's test and 93.9% had abnormal sialometry).

**Table 1. Demographic, laboratory and treatment characteristics of pSS cohort according to disease activity measured by ESSDAI.**

| | All patients | ESSDAI <5 | ESSDAI 5 -≤13 | ESSDAI >14 | P-value |
|---|---|---|---|---|---|
| | n = 49 | n = 23 | n = 18 | n = 8 | |
| **Demografics** mean ± SD | | | | | |
| Age at visit [years] | 61.2 ± 10.1 | 61.5 ± 9.6 | 60.0 ± 10.4 | 63.0 ± 11.5 | 0.78 |
| Age at diagnosis [years] | 53.3 ± 11.2 | 54.0 ± 11.5 | 52.5 ± 10.8 | 53.1 ± 12.2 | 0.91 |
| Disease duration [years] | 7.9 ± 8.4 | 7.6 ± 8.0 | 7.5 ± 7.4 | 9.8 ± 12.3 | 0.91 |
| Sex, n = females (%) | 47 (95.9) | 22 (95.7) | 17 (94.4) | 8 (100) | 0.80 |
| Smoking, n = smokers (%) | 3 (6.1) | 2 (8.7) | 1 (5.6) | 0 | 0.67 |
| BMI [kg/m$^2$] | 25.4 ± 5.6 | 27.0 ± 5.6 | 25.7 ± 5.5 | 20.2 ± 2.1 | **0.004** |
| **Biochemical assessments** mean ± SD | | | | | |
| *Autoantibodies* | | | | | |
| ANA pos, n (%) | 42 (85.7) | 22 (95.7) | 14 (77.8) | 6 (75.0) | 0.17 |
| ANA titer [AU] | 1026 ± 426 | 998 ± 466 | 972 ± 425 | 1280 ± 0 | 0.33 |
| Anti-Ro52 pos, n (%) | 31 (63.3) | 15 (65.2) | 10 (55.6) | 6 (75.0) | 0.62 |
| Anti-Ro52 titer [AU] | 1126 ± 683 | 1034 ± 669 | 930 ± 769 | 1685 ± 0 | **0.05** |
| Anti-Ro60 pos, n (%) | 30 (61.2) | 15 (65.2) | 9 (50.0) | 6 (75.0) | 0.42 |
| Anti-Ro60 titer [AU] | 1221 ± 362 | 1105 ± 482 | 1311 ± 127 | 1375 ± 0 | 0.36 |
| Anti-La pos, n (%) | 23 (46.9) | 10 (43.5) | 7 (38.9) | 6 (75.0) | 0.21 |
| Anti-La titer [AU] | 754 ± 633 | 552 ± 583 | 885 ± 675 | 939 ± 675 | 0.38 |
| dsDNA pos, n (%) | 12 (24.5) | 6 (26.1) | 5 (27.8) | 1 (12.5) | 0.68 |
| IgM-RF pos, n (%) | 33 (67.3) | 17 (73.9) | 10 (55.6) | 6 (75.0) | 0.41 |
| IgM-RF titer [IU/mL] | 139 ± 120 | 130 ± 116 | 164 ± 126 | 122 ± 138 | 0.58 |
| *Immunoglobulins* | | | | | |
| IgA [g/L] | 2.60 ± 1.20 | 2.74 ± 1.28 | 2.59 ± 1.04 | 2.21 ± 1.37 | 0.57 |
| IgG [g/L] | 14.51±5.63 | 13.39±4.16 | 14.73± 4.57 | 17.23±8.80 | 0.54 |
| IgM [g/L] | 1.28 ± 1.31 | 1.43±1.79 | 1.07±0.61 | 1.34±0.76 | 0.55 |
| *Complement* | | | | | |
| C3c [g/L] | 1.08± 0.20 | 1.10 ± 0.19 | 1.11 ± 0.21 | 0.93 ± 0.10 | 0.06 |
| C3d [g/L] | 56.5 ± 21.6 | 56.5 ± 22.5 | 57.0 ± 22.8 | 55.3 ± 18.5 | 1 |
| C4c [g/L] | 0.19 ± 0.07 | 0.20 ± 0.07 | 0.20 ± 0.07 | 0.15 ± 0.07 | 0.16 |
| CRP [mg/dl] | 3.1 ± 3.6 | 3.4 ± 4.1 | 3.0 ± 3.1 | 2.1 ± 3.4 | 0.41 |
| Cryoglobulin pos, n (%) | 10 (21.7) | 3 (13.0) | 5 (27.8) | 2 (25.0) | 0.62 |
| **Treatment,** mean ± SD | | | | | |
| Prednisolone, n (%) | 6 (12.2) | 0 | 5 (27.8) | 1 (12.5) | |
| Prednisolone [mg/day] | 4.2 ± 3.0 | 0 | 3.0 ± 1.1 | 10 | |
| Methotrexate, n (%) | 6 (12.2) | 4 (17.4) | 2 (11.1) | 0 | |
| Methotrexate [mg/week] | 17.1 ± 5.1 | 16.9 ± 6.3 | 17.5 ± 3.5 | 0 | |
| Hydroxychloroquine, n (%) | 21 (42.9) | 10 (43.5) | 8 (44.4) | 3 (37.5) | |
| Azathioprine, n (%) | 1 (2.0) | 0 | 1 (5.5) | 0 | |
| Mycophenolate mofetil, n (%) | 2 (4.1) | 0 | 1 (5.5) | 1 (12.5) | |
| Rituximab, n (%) | 1 (2.0) | 0 | 0 | 1 (12.5) | |
| **ESSPRI,** mean±SD | | | | | |
| Dryness (0–10) | 7.4 ± 2.1 | 7.1 ± 2.3 | 7.2 ± 1.9 | 8.5 ± 1.3 | 0.24 |
| Fatigue (0–10) | 7.0 ± 2.1 | 6.2 ± 2.6 | 7.7 ± 1.1 | 7.9 ± 1.6 | 0.06 |
| Pain (0–10) | 5.8 ± 2.9 | 5.7 ± 2.7 | 5.7 ± 3.2 | 6.3 ± 3.1 | 0.76 |

*(Continued)*

**Table 1.** (Continued)

| | All patients | ESSDAI <5 | ESSDAI 5 -≤13 | ESSDAI >14 | P-value |
|---|---|---|---|---|---|
| | n = 49 | n = 23 | n = 18 | n = 8 | |
| Total ESSPRI | 6.8 ± 1.8 | 6.7 ± 2.0 | 6.7 ± 1.6 | 7.8 ± 1.3 | 0.29 |

ESSDAI, the EULAR Sjögren's syndrome disease activity index; BMI, body mass index; ANA, anti-nuclear antibodies; ds-DNA, double stringed DNA; IgM RF, Immunoglobulin M rheuma-factor; CRP, C-reactive protein, ESSPRI, the EULAR Sjögren's Syndrome Patient Reported Index. Cryoglobulin was only measured in 46 patients

Normal reference

Autoantibodies ANA titer <80 AU, Anti-Ro52 titer < 20 AU, Anti-Ro60 titer < 20AU, Anti-La titer < 20 AU, IgM RF < 15 IU/ml. Immunoglobulins Ig A 0.7–4.30 g/ L, Ig G 6.1–15.7 g/L, Ig M 0.4–2.3 g/L. Complements C3 0.9–1.80 g/L, C4 0.10–0.4 g/L, C3d 20–52 g/l. CRP < 6 mg/dl

## Discussion

When using the OMERACT SGUS definitions and scoring system, abnormal salivary gland parenchyma in at least one gland is associated with the presence of several pSS related autoantibodies: ANA, Anti-Ro52, Anti-Ro60, Anti-La and IgM-RF, and in particular to high titer ANA and anti-Ro52. No correlation was found between abnormal SGUS and ESSDAI- or ESSPRI-scores.

Our cohort, like others, had female predominance. The patients were mean 53.3 years at diagnosis and mean 61.2 years at the assessment compared to mean age at the assessment of 58 years and mean disease duration of 5 years in a corresponding cohort by Seror et al. [22]. On the other hand, the present cohort had a slightly higher mean ESSDAI score of 7.2 (SD 6.4) as compared to 3.03 (SEM 0.31) among Korean patients, mean 2.8 (SD 4.1) among Turkish patients and median 2 (IQR 0–7) among French patients [22–24]. Moreover, the Danish patients self-reported slightly higher ESSPRI sum-scores of mean 6.8 (SD 1.8) as compared to Korean mean 4.65 (SD 2.3), Turkish mean 4.9 (SD 2.2) and French patients' median 5.7 (IQR 4–7).

**Table 2. OMERACT ultrasound score distribution in pSS cohort in one or two glands.**

| | Patients |
|---|---|
| | n (%) |
| **Highest US score in 1 gland** | |
| All glands | |
| ≥ 1 | 39 (79.6) |
| ≥ 2 | 26 (53.1) |
| = 3 | 12 (24.5) |
| Parotid gland | |
| ≥ 1 | 30 (61.2) |
| ≥ 2 | 25 (51.0) |
| = 3 | 9 (18.4) |
| Submandibular gland | |
| ≥ 1 | 39 (79.6) |
| ≥ 2 | 24 (49.0) |
| = 3 | 10 (20.4) |
| **At least 2 glands with US score** | |
| ≥ 1 | 38 (77.6) |
| ≥ 2 | 25 (51.0) |
| = 3 | 10 (20.4) |

**Table 3. Univariate analysis comparing pSS patients' characteristics with pathological SGUS in one or more glands versus non gland pathology.**

|  | All patients | Pathology ≥ 1 glands | No gland pathology | P-value |
|---|---|---|---|---|
|  | **n = 49** | **n = 26** | **n = 23** |  |
| ESSDAI | 7.2 ± 6.4 | 7.7 ± 7.6 | 6.6 ± 4.8 | 0.95 |
| ESSPRI | 6.8 ± 1.8 | 6.8 ± 1.9 | 6.9 ± 1.7 | 0.72 |
| Abnormal Schirmer's test, n(%) | 43 (87.8) | 22 (84.6) | 21 (91.3) | 0.48 |
| Abnomal sialometry, n(%) | 46 (93.9) | 25 (96.2) | 21 (91.3) | 0.48 |
| *Autoantibodies* |  |  |  |  |
| ANA pos, n(%) | 42 (85.7) | 26 (100) | 16 (69.6) | **0.002** |
| ANA titer [U] | 1026 ± 426 | 1148 ± 350 | 850 ± 474 | **0.02** |
| Anti-Ro52 pos, n(%) | 31 (63.3) | 22 (84.6) | 9 (39.1) | **0.001** |
| Anti-Ro52 titer [U] | 1126 ± 683 | 1289 ± 610 | 728 ± 720 | **0.03** |
| Anti-Ro60 pos, n(%) | 30 (61.2) | 22 (84.6) | 8 (34.8) | **0.000** |
| Anti-Ro60 titer [U] | 1221 ± 362 | 1214 ± 395 | 1241 ± 275 | 0.78 |
| Anti-La pos, n(%) | 23 (46.9) | 19 (73.1) | 4 (17.4) | **0.000** |
| Anti-La titer [U] | 754 ± 633 | 743 ± 605 | 808 ± 858 | 0.93 |
| IgM-RF pos, n(%) | 33 (67.3) | 24 (92.3) | 9 (39.1) | **0.000** |
| IgM-RF titer [IU/mL] | 139 ± 120 | 149 ± 122 | 112 ± 120 | 0.16 |

Furthermore, autoantibodies including ANA and IgM-RF occurred with high frequencies among patients in this study cohort (ANA 85.7%, IgM-RF 67.3%) as compared to the study cohort described by Cho et al. (ANA: 61.6%, IgM-RF: 47%) [23]. Also anti-Ro (63.3%) and anti-La/SSB (46.9%) autoantibodies were frequent in the Danish cohort compared with the study cohorts described by Inanc et al (anti-Ro: 46%, anti-La: 22%) [24] and Seror et al (anti-Ro: 59.2%, anti-La: 33.5%) [22].

In this study cohort, when using the validated, consensus-based OMERACT scoring system significantly more patients with abnormal SGUS, defined as at least one gland with positive ultrasound signs grade 2 or 3, had autoantibody positivity. The titers of ANA and anti-Ro52 were significantly higher in patients with abnormal SGUS. Several studies have described similar findings though using different SGUS scoring systems and definitions: Lee et al. found that "double seropositivity" of anti-Ro and anti-La were independent predictive factors of structural damage visualized by ultrasound [25]. Mossel et al. found that the SGUS score was significantly higher in patients with anti-Ro than patients without anti-Ro [9]. Wernicke et al. found that inhomogeneity of the salivary glands visualized by SGUS was associated with ANA, anti-Ro and anti-La positivity and that parotid inhomogeneity was more frequently detected in patients with double positivity of anti-Ro and anti-La [26]. Zandonella Calleger et al identified different phenotypes of pSS by SGUS, and normal appearing glands were associated with the absence of anti-La [27]. Interestingly, a study found that anti-Ro52 was highly expressed in serum and saliva samples of pSS patients and that the degree of ductal epithelial expression of anti-Ro52 correlated with the level of inflammation [28].

Our findings suggest that the autoantibodies detected in the blood samples have a strong association with the salivary glandular pathology detected by SGUS, probably as part of local inflammation and structural damage of the salivary glands according to Reed et al. [2]. The pathogenesis of pSS is multifactorial with important contributions from genetic susceptibility, infection and local inflammation [29]. The B-cells are central in this process linking local inflammation ("overactive" B-cells) in the salivary glands with the systemic synthesis of autoantibodies [30].

Only half of the patients with pathological Schirmer's and sialometry had normal SGUS. This is in line with an ultrasound study of patients referred with dryness syndrome on suspicion of pSS where 49% of the patients diagnosed pSS had SGUS at 0 or 1 [17].

No difference was found in the number of patients classified with abnormal SGUS independent of applying cut off with at least one or at least two glands with US score > 1, leading to equal sensitivity of both evaluations methods. And the distribution of diseased glands according to the definition SGUS was distributed between the PG and the SMG.

Previous studies using different US scoring systems for gland pathology have found an association between ESSDAI and SGUS, but in this cross-sectional study, the total ESSDAI score was not correlated with OMERACT SGUS scores [31, 32].

One explanation for the lack of an association between SGUS and ESSDAI could be that the ESSDAI contains measures of both disease activity and potential measures of tissue damage, and these dimensions may not be strongly associated with salivary gland involvement. The SGUS scoring system is based on structural damage in the salivary glands and is not a fluctuating measurement as the ESSDAI. Correspondingly, SGUS findings did not reflect the patient reported ESSPRI where extra glandular symptoms account for most domains.

The limitations of this single-centre, cross-sectional study makes the causality of observed correlations impossible to investigate. pSS is a disease with a long diagnostic delay, and the cohort consisted of patients with different medical treatment and observation time. In addition, the relatively small number of patients decreases the statistical power.

The strengths of this study include a well-characterized study cohort that is fulfilling the 2002 AECG diagnostic criteria. Furthermore, we applied the OMERACT validated ultrasound definitions and scoring system for SG pathology and all examinations were performed by one experienced ultrasonographer ensuring reliable measures of SGUS.

In conclusion, we found an association between SGUS and the presence of pSS related autoantibodies, including ANA, anti-Ro52, anti-Ro60, anti-La and IgM-RF in a cohort of Danish pSS patients using the OMERACT ultrasound definition and scoring system for SG. The association might indicate an expression of autoimmunity in salivary glands and a potential pathogenic link [29, 30]. This study discloses the high value of SGUS to assess the disease involvement of salivary glands. The lack of correlation between scores of SGUS and disease activity measures such as ESSDAI and ESSPRI might indicate that these also describe other aspects of the disease than mere gland pathology.

## Supporting information

**S1 File.**
(PDF)

## Acknowledgments

Study data were collected and managed using REDCap electronic data capture tools hosted at OPEN, Open Patient data Explorative Network, Odense University Hospital, Region of Southern Denmark [33, 34].

## Author Contributions

**Conceptualization:** Nanna Surlemont Schmidt, Anne Voss, Hanne M. Lindegaard.

**Data curation:** Nanna Surlemont Schmidt, Lene Terslev, Hanne M. Lindegaard.

**Formal analysis:** Nanna Surlemont Schmidt, Anne Voss, Anna Christine Nilsson, Hanne M. Lindegaard.

**Funding acquisition:** Nanna Surlemont Schmidt, Hanne M. Lindegaard.

**Investigation:** Nanna Surlemont Schmidt, Søren Andreas Just.

**Methodology:** Nanna Surlemont Schmidt, Anne Voss, Lene Terslev, Hanne M. Lindegaard.

**Project administration:** Nanna Surlemont Schmidt.

**Resources:** Nanna Surlemont Schmidt.

**Supervision:** Anne Voss, Anna Christine Nilsson, Lene Terslev, Hanne M. Lindegaard.

**Validation:** Nanna Surlemont Schmidt, Lene Terslev.

**Writing – original draft:** Nanna Surlemont Schmidt, Anne Voss, Anna Christine Nilsson, Lene Terslev, Søren Andreas Just, Hanne M. Lindegaard.

**Writing – review & editing:** Nanna Surlemont Schmidt, Anne Voss, Anna Christine Nilsson, Lene Terslev, Søren Andreas Just, Hanne M. Lindegaard.

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
