## [Decision Letter · Decision Letter 0]

3 Dec 2021

PONE-D-21-25023A Danish single-center study of patients with Sjögren’s Syndrome: Pathologic salivary gland ultrasound is associated with the presence of autoantibodiesPLOS ONE

Dear Dr. Lindegaard,

Thank you for submitting your manuscript to PLOS ONE. After careful consideration, we feel that it has merit but does not fully meet PLOS ONE’s publication criteria as it currently stands. Therefore, we invite you to submit a revised version of the manuscript that addresses the points raised by the revierews during the review process.

We look forward to receiving your revised manuscript.

Kind regards,

Silke Appel, PhD (Dr. rer. nat.)

Academic Editor

PLOS ONE

https://journals.plos.org/plosone/s/file?id=ba62/PLOSOne_formatting_sample_title_authors_affiliations.pdf"

2. Please note that all PLOS journals ask authors to adhere to our policies for sharing of data and materials: https://journals.plos.org/plosone/s/data-availability. According to PLOS ONE’s Data Availability policy, we require that the minimal dataset underlying results reported in the submission must be made immediately and freely available at the time of publication. As such, please remove any instances of 'unpublished data' or 'data not shown' in your manuscript and replace these with either the relevant data (in the form of additional figures, tables or descriptive text, as appropriate), a citation to where the data can be found, or remove altogether any statements supported by data not presented in the manuscript.

In your Methods section, please provide a justification for the sample size used in your study, including any relevant power calculations (if applicable).

3. Please provide additional details regarding participant consent. In the ethics statement in the Methods and online submission information, please ensure that you have specified whether consent was written or verbal/oral. If consent was verbal/oral, please specify: 1) whether the ethics committee approved the verbal/oral consent procedure, 2) why written consent could not be obtained, and 3) how verbal/oral consent was recorded. If your study included minors, please state whether you obtained consent from parents or guardians in these cases. If the need for consent was waived by the ethics committee, please include this information.

5. Thank you for stating the following in the Funding Section of your manuscript:

“The study was supported by The Danish Rheumatism Association.”

“NNS

The study was supported by The Danish Rheumatism Association.”

Reviewers' comments:

Reviewer's Responses to Questions

**Comments to the Author**

1. Is the manuscript technically sound, and do the data support the conclusions?

Reviewer #1: Yes

Reviewer #2: Yes

2. Has the statistical analysis been performed appropriately and rigorously? 

Reviewer #1: Yes

Reviewer #2: Yes

3. Have the authors made all data underlying the findings in their manuscript fully available?

Reviewer #1: Yes

Reviewer #2: Yes

4. Is the manuscript presented in an intelligible fashion and written in standard English?

Reviewer #1: Yes

Reviewer #2: Yes

5. Review Comments to the Author

Reviewer #1: 1- I suggest that the title is revised in the interest of clarity to:

Salivary gland ultrasound is associated with the presence of autoantibodies in patients with Sjögren’s Syndrome: A Danish single-centre study

2- Line 92: Explain “focus score” as this is the first time it is mentioned in this manuscript. A focus score of 1 or greater is diagnostic of Sjogren’s and not as stated in line 178 (…a focus score of > 1..)

3- Swap the numbering of table 2 and table 3, as table 3 is referred to in line 207 before table 3 in line 216

Reviewer #2: This is a comprehensive and well conducted study aiming to investigate whether ultrasound findings of major salivary glands in Sjögren’s syndrome are correlated with serological markers, autoantibodies, patient- or doctor-reported disease activity in a Danish cohort. One of the strengths of the study is that only one “ultrasounder” was used. The following conclusions could be drawn: Abnormal ultrasound findings are associated with autoantibodies of high specificity for Sjögren’s but not with ESSDAI, ESSPRI or inflammatory markers.

The following comments and questions can be raised:

1. Disease duration – from the time of diagnosis to the SGUS analyses. I think this should be elaborated on further and in particular if the results are related to a short disease duration as well as if younger versus older patients influenced the SGUS data.

2. The outcome measures used (and these are currently the only available) might not be the most sensitive/optimal. This certainly makes it difficult to relate the SGUS data to. Maybe discuss this further?

3. One can assume that the most severe SGUS data recorded is a result of a longstanding chronic inflammation. Are there any studies that you refer to that brings this up? Maybe also discuss this further?

4. Autoantibodies are known to occur years before symptoms onset and diagnosis in other rheumatic diseases (RA, SLE) but also in Sjögren’s. Do you dare to speculate on this in relation to your SGUS results?

5. The DISCUSSION could be made more exciting for the reader as well as shortened. Have you selected the most novel and interesting data and discussed these? Is all the repetition of results necessary?

6. PLOS authors have the option to publish the peer review history of their article (what does this mean?). If published, this will include your full peer review and any attached files.

Reviewer #1: No

Reviewer #2: No

---

## [Author Response · Author response to Decision Letter 0]

16 Jan 2022

We have covered all the questions in the decision letter in the file letter to editor 13.12.21 (named response to reviewers)

---

## [Decision Letter · Decision Letter 1]

23 Feb 2022

Salivary gland ultrasound is associated with the presence of autoantibodies in patients with Sjögren´s Syndrome: A Danish single-centre study

PONE-D-21-25023R1

Dear Dr. Lindegaard,

We’re pleased to inform you that your manuscript has been judged scientifically suitable for publication and will be formally accepted for publication once it meets all outstanding technical requirements.

Kind regards,

Silke Appel, PhD (Dr. rer. nat.)

Academic Editor

PLOS ONE

Additional Editor Comments (optional):

Reviewers' comments:

Reviewer's Responses to Questions

**Comments to the Author**

1. If the authors have adequately addressed your comments raised in a previous round of review and you feel that this manuscript is now acceptable for publication, you may indicate that here to bypass the “Comments to the Author” section, enter your conflict of interest statement in the “Confidential to Editor” section, and submit your "Accept" recommendation.

Reviewer #1: All comments have been addressed

Reviewer #2: All comments have been addressed

2. Is the manuscript technically sound, and do the data support the conclusions?

Reviewer #1: Yes

Reviewer #2: Yes

3. Has the statistical analysis been performed appropriately and rigorously? 

Reviewer #1: Yes

Reviewer #2: Yes

4. Have the authors made all data underlying the findings in their manuscript fully available?

Reviewer #1: Yes

Reviewer #2: Yes

5. Is the manuscript presented in an intelligible fashion and written in standard English?

Reviewer #1: Yes

Reviewer #2: Yes

6. Review Comments to the Author

Reviewer #1: (No Response)

Reviewer #2: Interesting study on a relative novel method to analyze inflammation in major salivary glands. The authors have responded appropriately to my comments. I have no further comments.

7. PLOS authors have the option to publish the peer review history of their article (what does this mean?). If published, this will include your full peer review and any attached files.

Reviewer #1: No

Reviewer #2: No

---

## [Editor Report · Acceptance letter]

10 Mar 2022

PONE-D-21-25023R1 

Salivary gland ultrasound is associated with the presence of autoantibodies in patients with Sjögren’s Syndrome: A Danish single-centre study

Dear Dr. Lindegaard:

I'm pleased to inform you that your manuscript has been deemed suitable for publication in PLOS ONE. Congratulations! Your manuscript is now with our production department. 

Kind regards, 

on behalf of

Dr. Silke Appel 

Academic Editor

PLOS ONE